# Tissue inflammation induced by constitutively active STING is mediated by enhanced TNF signaling

Hella Luksch[1], Felix Schulze[1], David Geißler-Lösch[2], David Sprott[3], Lennart Höfs[2], Eva M Szegö[2], Wulf Tonnus[4], Stefan Winkler[1], Claudia Günther[5], Andreas Linkermann[4], Rayk Behrendt[6], Lino L Teichmann[7], Björn H Falkenburger[2,8], Angela Rösen-Wolff[1]*

[1]Department of Pediatrics, Faculty of Medicine and University Hospital Carl Gustav Carus, Technische Universität Dresden, Dresden, Germany; [2]Department of Neurology, Faculty of Medicine and University Hospital Carl Gustav Carus, Technische Universität Dresden, Dresden, Germany; [3]Department of Physiology, Faculty of Medicine and University Hospital Carl Gustav Carus, Technische Universität Dresden, Dresden, Germany; [4]Division of Nephrology, Department of Internal Medicine III, Faculty of Medicine and University Hospital Carl Gustav, Dresden, Germany; [5]Department of Dermatology, Faculty of Medicine and University Hospital Carl Gustav Carus, Technische Universität Dresden, Dresden, Germany; [6]Institute for Clinical Chemistry and Clinical Pharmacology, University Hospital Bonn, Bonn, Germany; [7]Department of Medicine III, University Hospital Bonn, Bonn, Germany; [8]Deutsches Zentrum für Neurodegenerative Erkrankungen, Dresden, Germany

*For correspondence:
angela.roesen-wolff@tu-dresden.de

Competing interest: The authors declare that no competing interests exist.

**Abstract** Constitutive activation of STING by gain-of-function mutations triggers manifestation of the systemic autoinflammatory disease STING-associated vasculopathy with onset in infancy (SAVI). In order to investigate the role of signaling by tumor necrosis factor (TNF) in SAVI, we used genetic inactivation of TNF receptors 1 and 2 in murine SAVI, which is characterized by T cell lymphopenia, inflammatory lung disease, and neurodegeneration. Genetic inactivation of TNFR1 and TNFR2, however, rescued the loss of thymocytes, reduced interstitial lung disease, and neurodegeneration. Furthermore, genetic inactivation of TNFR1 and TNFR2 blunted transcription of cytokines, chemokines, and adhesions proteins, which result from chronic STING activation in SAVI mice. In addition, increased transendothelial migration of neutrophils was ameliorated. Taken together, our results demonstrate a pivotal role of TNFR signaling in the pathogenesis of SAVI in mice and suggest that available TNFR antagonists could ameliorate SAVI in patients.

## Editor's evaluation

This study provides important insights into the pathogenesis of systemic autoinflammatory disease STING-associated vasculopathy with onset in infancy (SAVI). Specifically, the authors demonstrate the critical role of tumor necrosis factor (TNF) signaling in SAVI in mouse model. Overall, the data presented are convincing, supporting the author's conclusions.

## Introduction

Stimulator of interferon response cGAMP interactor 1 (STING) is a key regulator in innate immunity, especially in the defense against viral infections. Uncontrolled activity of STING results in manifestation of autoinflammatory diseases, e.g., STING-associated vasculopathy with onset in infancy (SAVI) (*Liu et al., 2014*), Parkinson's disease (*Hinkle et al., 2022*), or severe COVID-19 disease (*Domizio et al., 2022*). Murine SAVI is a well-established model for pathology and signaling of constitutive uncontrolled STING activation (*Warner et al., 2017*). STING is an innate immune receptor that senses cyclic dinucleotides. These can be derived from bacteria or in mammalian cells be produced by the enzyme cyclic GMP-AMP synthase (cGAS), which is activated upon binding to double-stranded DNA. Mammalian cGAS produces the unique 2′3′-cGAMP that binds to STING and results in its translocation, phosphorylation, and oligomerization. STING oligomers recruit TANK-binding kinase 1 (TBK1), which subsequently activates type I interferon (IFN I) signaling as well as nuclear factor κ light chain enhancer of activated B cells (NF-κB) triggered signaling (*Balka et al., 2020*; *de Oliveira Mann et al., 2019*; *Hopfner and Hornung, 2020*). STING signaling is terminated by AP-1-mediated sorting of phosphorylated STING into clathrin-coated transport vesicles that fuse with endolysosomes to degrade STING (*Gonugunta et al., 2017*; *Liu et al., 2022a*).

Uncontrolled activity of STING is associated with various autoinflammatory disorders and severe diseases caused by viral infections (*Deng et al., 2020*; *Yang et al., 2022*). Gain-of-function mutations of human *STING1* cause SAVI, which is characterized by severe interstitial lung disease, T cell lymphopenia, skin inflammation, and perturbed IFN- and NF-κB-driven signaling (*Clarke et al., 2020*; *Liu et al., 2014*; *Picard et al., 2016*; *Tang et al., 2020*). Genetically induced chronic activation of STING results in comparable severe systemic autoinflammatory symptoms in the murine organism (*Bennion et al., 2019*; *Bennion et al., 2020*; *Gao et al., 2022*; *Luksch et al., 2019*; *MacLauchlan et al., 2023*; *Martin et al., 2019*; *Motwani et al., 2019*; *Platt et al., 2021*; *Shmuel-Galia et al., 2021*; *Siedel et al., 2020*; *Stinson et al., 2022*; *Szego et al., 2022*; *Warner et al., 2017*). We previously established a SAVI mouse model, by knocking in the disease causing variant N153S into the endogenous murine *Sting1* gene (STING ki) resulting in T cell lymphopenia, interstitial lung disease, and systemic autoinflammation (*Luksch et al., 2019*). In addition, STING ki mice show degeneration of dopaminergic neurons induced by neuroinflammation (*Szego et al., 2022*).

Initially, STING activates various signal transduction pathways and has been assumed to function primarily through type I IFN signaling (*Liu et al., 2014*). Yet, the manifestation of SAVI hallmarks in STING ki mice were unaffected by knockout of cGAS, IFNAR1, IRF3, and IRF7 (*Luksch et al., 2019*; *Siedel et al., 2020*), suggesting that other pathways in the STING signaling cascade are required for SAVI symptoms. In our previous investigations, we observed an elevated transcription of *Tnf* in spleen and thymus of STING ki mice (*Siedel et al., 2020*). In order to test the hypothesis that tumor necrosis factor (TNF) signaling is involved in manifestation and progression of murine SAVI disease, we here used genetic inhibition of TNF receptors.

TNF, a systemic multifunctional cytokine, is involved in inflammation and immune regulation as well as development of lymphoid organs, and TNF can be produced and secreted by nearly every cell. We established new mouse lines with STING ki in addition to genetic depletion of tumor necrosis factor receptor 1 (TNFR1), tumor necrosis factor receptor 2 (TNFR2), or TNFR1/2. Both receptors are involved in TNF signaling pathways with different functions. TNFR1 is constitutively expressed on almost all cell populations, whereas TNFR2 expression is inducible in specific cell types, e.g., immune and endothelial cells (*Wajant and Siegmund, 2019*). Both receptors are able to bind TNF but only TNFR1 contains an intracellular death domain and is involved in programmed cell death. TNFR2 is associated with survival and proliferation of cells (*Atretkhany et al., 2020*). After binding of ligands, TNF receptors trigger NF-κB-driven signaling, predominantly through TNFR1 whereas TNFR2 activates NF-κB transcription poorly (*McFarlane et al., 2002*). Current publications suggest distinct effects of TNFR1 and TNFR2 in association with different types of inflammation. TNFR1 signaling stimulates proinflammatory responses within the innate immune system. In contrast, actions of TNFR2 are involved in cellular homeostasis and anti-inflammatory responses (*Liang et al., 2022*). Moreover, in the murine model of autoinflammatory familial Mediterranean fever (FMF), both TNF receptors have opposite effects. TNFR1 showed a pathogenic role and TNFR2 a protective role in association with murine FMF (*Sharma et al., 2019*).

In this work, we observed that pathology in the thymus of SAVI mice was dependent on TNFR1 and TNFR2 signaling whereas signaling through TNFR1 but not on TNFR2-mediated pathology in lungs of STING ki mice. Similarly, the manifestation of dopaminergic neuron degeneration in substantia nigra was dependent on TNFR1/2 signaling. Finally, we investigated the role of constitutive STING activation on the endothelial barrier, which was found to be TNF signaling-dependent. Endothelial cells of STING ki mice induced transmigration of neutrophils in a TNFR1-dependent manner. Overall, our study highlights a pivotal role of TNF signaling in SAVI disease and implies TNF blockade as valuable therapeutic option to ameliorate symptoms of SAVI disease in patients.

## Materials and methods

### Animals

Heterozygous STING N153S/WT mice (STING ki) were previously described (*Luksch et al., 2019*). STING WT and STING ki mice were crossed to *Tnfr1/2*$^{-/-}$ mice, kindly provided by Hans-Joachim Anders, Munich, Germany (*Mulay et al., 2017*). All mice were housed at the Experimental Center of the University of Technology Dresden under specific pathogen-free conditions. All mice experiments were approved by the Landesdirektion Sachsen (TVV 4/2019, TVV 13/2019) and carried out in accordance with the institutional guidelines on animal welfare.

### Cell preparations and flow cytometry

Spleens and thymi were mechanically homogenized and passed through a 100 μm cell strainer. Single-cell suspensions of spleen, thymi, and blood were obtained by lysis of red blood cells and additional filtration through 30 μm meshes. All cell suspensions were washed with FACS buffer (PBS, 2%FCS, 2.5 mM EDTA). These isolated cells were incubated with fluorescence-labeled antibodies in FACS buffer for 30 min at 4°C. For a detailed overview of used antibodies, see *Supplementary file 1, table S1*. After incubation, cells were washed twice with FACS buffer. Exclusion of dead cells was performed by adding Zombie UV dye (BioLegend, USA). Cells were analyzed by LSR II and LSR FORTESSA (BD Biosciences, Germany) and evaluated with FlowJo V10 software (Tree Star, USA).

### Histology of lung

Lung tissue was dissected from mice and fixed in 4% formaldehyde for 24 hr at 4°C and embedded in paraffin. Lung tissue sections (thickness of 3 μm) were stained with Mayer's hemalum (Carl Roth, Germany) and counterstained with eosin (Carl Roth, Germany). For quantification of lung disease, whole tissue sections were scanned by Axio Scan Z1 and ZEN software (both from Zeiss, Germany). ZEN 3.0 (Zeiss, Germany) software was used for evaluation of inflammatory areas in all analyzed lung sections. The area of diseased lung in each section was calculated by inflamed area divided by total area of lung section, excluding large airway spaces.

### Dye labeling of lymph nodes

Identification of lymph nodes was performed as described previously (*Harrell et al., 2008*). Briefly, mice were anesthetized and 25 μl of 5% Evans Blue dye in PBS was injected subcutaneously into both hind paws. After 30 min of dye injection, mice were euthanized and dissected for visualization of blue-labeled lymph nodes.

### Gene expression analysis

Total RNA of lung and thymi was extracted from snap-frozen tissue by using the RNeasy Mini Kit (QIAGEN, Germany) according to the manufacturer's instructions and cDNA was generated by MMLV reverse transcription (Promega, Germany). Quantitative real-time PCR assays were carried out by using GoTaq qPCR master mix (Promega, Germany) and QuantStudio 5 (Thermo Fisher Scientific, USA). PCR primers were generated from the Primer Bank database, see *Supplementary file 3, table S3* (*Spandidos et al., 2010*). Expression of genes was normalized with respect to each housekeeping gene using ΔΔCt method for comparing relative expression.

### LEGENDplex assay

Mouse cytokine release syndrome panel LEGENDplex (BioLegend, USA) is a multiplex bead-based assay using the basic methodology of ELISA. Snap-frozen lung tissue was extracted by 1% NP-40

(Sigma-Aldrich, Germany) and cOmplete Protease Inhibitor Cocktail (Roche, Germany) in PBS. Collected serum and lung extracts from mice were incubated with bead-conjugated antibodies overnight at 4°C and permanent shaking. Content of chemokines/cytokines was quantified after washing and staining with biotinylated detection antibodies and phycoerythrin-bound streptavidin by using LSR II (BD Biosciences, Germany). Calculation of each chemokine/cytokine quantity was determined by using standard curves according to the manufacturer's instructions.

## Immunofluorescence staining of brain sections

Mice were euthanized with an overdose of isoflurane (Baxter, Belgium) and perfused transcardially with 4% paraformaldehyde (PFA) in Tris-buffered saline (TBS, pH 7.6). The tissue was left in 4% PFA for another 48 hr at 4°C. For cryoprotection, brains were incubated in 30% sucrose in TBS. They were snap-frozen at –55°C in isopentane and stored at –80°C. 30-µm-thick coronal brain sections were obtained using a Cryostat (Leica CM3050 S Biosystems, Germany).

Brain sections were rinsed in TBS three times for 10 min each. Afterward they were incubated for 1 hr at room temperature in a blocking buffer, which consisted of 10% donkey serum (BIOZOL Diagnostica Vertrieb GmbH, Germany), 0.2% Triton X-100 (Thermo Scientific, USA), and TBS. Incubation with primary antibody was performed by using sheep anti-TH, chicken anti-GFAP, and guinea pig anti-Iba1 (*Supplementary file 2, table S2*) at 4°C overnight. After three 10 min rinses with TBS, slices were incubated with fluorophore-conjugated secondary antibodies for 1 hr at room temperature Alexa Fluor 488-conjugated donkey anti-sheep, Alexa Fluor 647-conjugated donkey anti-chicken, CF555-conjugated donkey anti-guinea pig (*Supplementary file 2, table S2*). Hoechst was applied for nuclear counterstaining. Sections were mounted in Fluoromount-G (Invitrogen, USA).

## Quantification of dopaminergic neurons, astrocytes, and microglia in the SNc

For each animal, we stained and slide-scanned every fourth brain section. Images were acquired using a spinning disk confocal microscope and a ×20/0.8 objective. The system consists of a Zeiss Axio Observer.Z1 Inverted Microscope (Zeiss, Germany) supported by a Yokogawa CSU-X1 unit (Yokogawa Life Science, Tokyo). Each section was scanned at seven Z-levels with 1 µm intervals and projected according to the 'Orthogonal projection' and 'Stitching' function in Zeiss Zen 3.1 software (Zeiss, Germany). TH-positive neurons of both hemispheres in the *pars compacta* of the *substantia nigra* (SNc) were manually counted in Zeiss Zen. Based on the TH staining, the SNc was manually encircled. The numbers of Iba1-positive microglia and GFAP-positive astrocytes were also quantified manually in both hemispheres within the marked area. Next, each cell type's count was summed and multiplied by four since every fourth section was analyzed in order to represent the total number within the SNc. Counts of positive stained cells from STING WT and STING ki mice were normalized to the corresponding mean count in STING WT.

## Transcriptomic analysis of murine lung endothelial cells

Murine lung endothelial cells were isolated from perfused (10 U/ml heparin diluted in PBS) lung tissue. Single-cell suspension was obtained after digestion with 1 mg/ml Collagenase (Sigma-Aldrich, Germany), 3.5 mg/ml Dispase (Roche, Germany), and 25 µg/ml DNaseI (Roche, Germany) in native IMDM (Thermo Fisher Scientific, USA) over 45 min at 37°C. Collected cells were washed with PBS and passed through a 30 µm cell strainer. Lung endothelial cells were enriched by positive selection of CD31$^+$ cells with microbeads (Miltenyi Biotec), according to the manufacturer's protocol. All selected cells were stained with antibodies (all from BioLegend, USA) against CD45.2 (1:300), CD11b (1:500), Ter-119 (1:500), CD326 (1:500), CD31 (1:300), and separated by FACS Aria. Dead cells and cellular debris were excluded by PI staining. The designed gating strategy ensured the exclusion of leukocytes (CD45.2$^+$ and CD11b$^+$), erythrocytes (Ter-119$^+$), and epithelial cells (CD326$^+$). Total RNA was extracted by RNeasy Plus Mini Kit (QIAGEN, Germany) and poly-A enriched before library preparation using NEBNext Ultra II Directional RNA Library Prep Kit (NEB, USA). For each library, 30 mio single end reads were generated on an Illumina NovaSeq 6000. Reads were mapped to mouse genome GRCm39 followed by normalization, exploratory, and differential expression analysis using DESeq2 (*Love et al., 2014*). Data are deposited on GEO database, Accession no. GSE244062.

## Functional assays of murine lung endothelial cells

Isolation and culture of murine lung endothelial cells was performed as described previously (*Fehrenbach et al., 2009*). In brief, mice were perfused transcardially with 10 U/ml heparin (Carl Roth, Germany) diluted in PBS. Lung tissue was removed and digested with 1 mg/ml Collagenase (Sigma-Aldrich, Germany), 3.5 mg/ml Dispase (Roche, Germany) and 25 µg/ml DNaseI (Roche) in native IMDM (Thermo Fisher Scientific, USA) over 45 min at 37°C. The resulting cell suspension was filtered through a 30 µm cell strainer. The filtered cell suspension was washed in PBS and resuspended in complete IMDM culture medium. The extracted cells of lung tissue were plated into Attachment Factor Protein- (Life Technologies, USA) coated T75 tissue culture flasks for 2 days. After this time, lung cells were detached by Accutase (BioLegend, USA) and the endothelial cells were separated using Dynabeads coupled to anti-CD31 antibody and anti-ICAM2 antibody (BioLegend, USA) according to the manufacturer's instructions (Thermo Fisher Scientific, USA). Collected murine lung endothelial cells were cultured in coated T75 flasks. Cultured cells from passages 1–3 were used for analyzing neutrophil attachment and neutrophil transendothelial migration.

For this purpose, $0.3\times10^5$ lung endothelial cells were seeded into 6-channel µ-Slides VI 0.4 (IBIDI, Germany), which were precoated with attachment factor (Thermo Fisher Scientific, USA). Cells were incubated for 4 days with changing media twice per day. For stimulated cells, last medium exchange was performed with either 5 ng/ml TNF (Peprotech, USA) or 100 ng/ml cLPS (Invivogen, USA) and incubation lasted overnight. For the migration assay, neutrophils were isolated from murine bone marrow using femur and tibia. Neutrophils were separated by gradient centrifugation using Histopaque 1119 and 1077 (Sigma-Aldrich, Germany) according to the manufacturer's instructions. Neutrophils were washed, counted, and diluted to $1\times10^6$ cells/ml. Afterward flow was applied to µ slides at a flow rate of 0.5 ml/min (≡0.6 dyn/cm$^2$) using a syringe pump for 10 min. Subsequently, $0.6\times10^6$ neutrophils were injected upstream of the endothelial monolayer via a port. Phase-contrast images (five per each channel) were taken with an AXIO OBSERVER Z1 (Zeiss, Germany) over 20 min at a speed of one image every 10 s. Evaluation of attached and transmigrated cells was performed using ZEN Blue software (Zeiss, Germany).

## Statistics

All statistical analyses were performed by using GaphPad Prism 9. In the graphs, markers represent data from individual animals and lines represent means of all mice from the indicated genotype. Grubbs' test was used to identify outliers. Comparison of two groups was performed by using Mann-Whitney test. For the comparison of more groups, one-way ANOVA including Dunnett's multiple comparisons test or Kruskal-Wallis test including Dunn's multiple comparisons test was used. Significance levels in each figure are indicated by symbols with *p≤0.05, **p<0.005, ***p<0.001, ****p<0.0001. Analysis of effect size and power were performed by G*Power 3.1.9.4 and collected in *Supplementary file 4, table S4*, effect size d convention (<0.5 = small, 0.5–0.8=medium, >0.8 = large effect) and effect size f convention (<0.25 = small, 0.25–0.4=medium, >0.4 = large effect).

## Results

### Knockout of TNFR1 and TNFR2 does not significantly affect the numbers of blood T cell in STING ki mice

We established the STING ki;*Tnfr1/2* $^{-/-}$ mouse line with the combination of the gain-of-function mutation of STING and nonfunctional TNFR1 and TNFR2 as double knockout (TNFR1/2). In addition, we generated two new mouse lines, STING ki;*Tnfr1*$^{-/-}$ (lacking TNFR1) and STING ki;*Tnfr2*$^{-/-}$ (lacking TNFR2). At the age of 10 weeks, mice were sacrificed and analyzed extensively in comparison to STING WT and STING ki mice (*Figure 1—figure supplement 1* for STING WT and *Figure 1* for STING ki). The genetic deletion of TNFR1, TNFR2 alone or together did not increase body weight in STING ki mice significantly (*Figure 1A*). Lack of TNFR1 resulted in elevation of blood CD4$^+$ T cell and CD8$^+$ T cell numbers in STING ki;*Tnfr1*$^{-/-}$ and STING ki;*Tnfr1/2* $^{-/-}$ mice (*Figure 1B–D*, large effect size f, *Supplementary file 4, table S4*). The frequency of blood CD4$^+$ naïve T cells (CD62L$^{hi}$, CD44$^{low}$) was increased in STING ki;*Tnfr1*$^{-/-}$ and STING ki;*Tnfr1/2*$^{-/-}$ mice in comparison to STING ki mice (*Figure 1E*, large effect size f, *Supplementary file 4, table S4*). The absence of TNFR1 or TNFR2 had no protective effect on naïve CD8$^+$ T cells (*Figure 1F*) and all effector T cell (CD62L$^{low}$, CD44$^{hi}$) populations in the

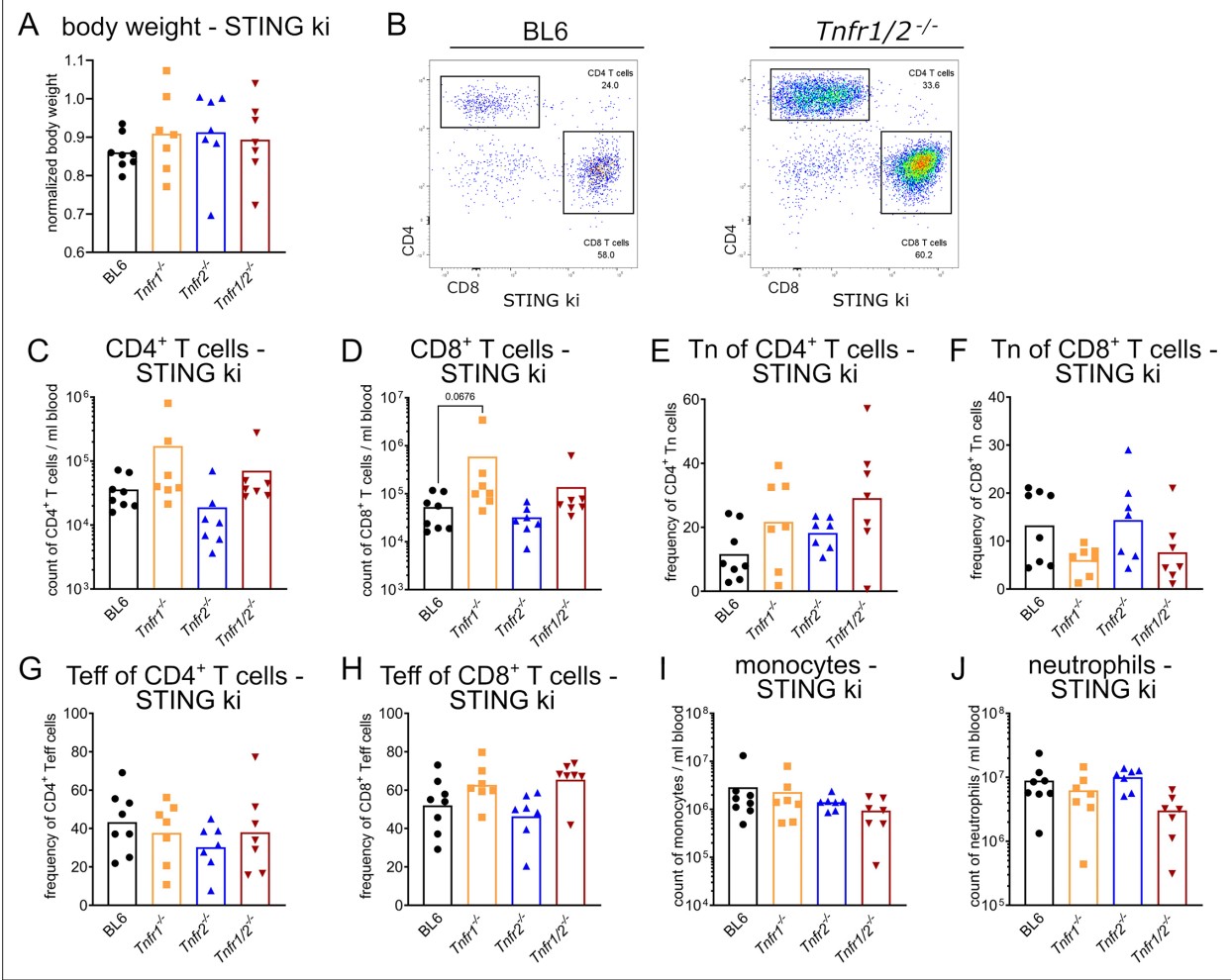

**Figure 1.** Disruption of TNFR signaling did not significantly prevent T cell lymphopenia in blood of STING ki mice. (**A**) Normalized body weight of 10-week-old STING ki mice, compared to body weight data from strain C57BL/6NJ (#005304, Jackson Laboratory). (**B**) Representative FACS plots of blood CD4+ T cells and CD8+ T cells from STING ki mice on C57BL/6 (BL6) or *Tnfr1/2-/-* background. (**C**) Numbers of blood CD4+ T cells in STING ki mice of indicated genotype. (**D**) Numbers of blood CD8+ T cells in STING ki mice of indicated genotype. (**E**) Frequency of blood naïve (Tn) CD4+ T cell population in STING ki mice of indicated genotype. (**F**) Frequency of blood naïve (Tn) T cells of CD8+ T cell population in STING ki mice of indicated genotype. (**G**) Frequency of blood effector (Teff) CD4+ T cell population in STING ki mice of indicated genotypes. (**H**) Frequency of blood effector (Teff) CD8+ T cell population in STING ki mice of indicated genotypes. (**I**) Numbers of blood monocytes in STING ki mice of indicated genotypes. (**J**) Numbers of blood neutrophils in STING ki mice of indicated genotypes. Markers represent individual mice, bars represent mean of n=7–8 mice per group pooled from nine independent preparations analyzed by Kruskal-Wallis test including Dunn's multiple comparisons test.

The online version of this article includes the following figure supplement(s) for figure 1:

**Figure supplement 1.** Disruption of TNFR signaling did not change the T cell numbers in the blood of STING WT mice.

blood of all STING ki mice (*Figure 1G and H*). In addition, blood myeloid cell numbers were reduced between STING ki with or without TNFR knockout (*Figure 1I and J*, large effect size f, *Supplementary file 4, table S4*).

Parallel characterization of these parameters in all STING WT mice (*Figure 1—figure supplement 1A–J*) showed no differences with exception of significantly more CD8+ T effector cells in the blood of TNFR1/2 ko mice (*Figure 1—figure supplement 1H*).

## Lack of TNFRs partially rescues thymus and spleen pathology in STING ki mice

The thymus undergoes massive pathological modifications in murine SAVI disease. As previously shown, the numbers of thymocytes were significantly reduced in STING ki mice. Likewise, various

inflammatory signaling pathways, detected by gene expression analysis of *Sting1*, *Cxcl10,* and *Tnf*, were upregulated. Finally, these inflammatory processes led to significant functional limitations of T cell maturation in the thymus of STING ki mice (*Siedel et al., 2020*).

In order to study thymus and spleen pathology here, TNFR knockout mice were sacrificed and analyzed in comparison to respective STING WT and STING ki mice (*Figure 2—figure supplement 1* for STING WT and *Figure 2* for STING ki) at the age of 10 weeks.

Knockout of TNFR1, 2 or double knockout of 1 and 2 (1/2) was first studied in comparison to STING WT;BL6 mice. As shown in *Figure 2—figure supplement 1*, knockout of TNFR signaling resulted in unaltered thymic cellular content (*Figure 2—figure supplement 1A*). Double negative cells were slightly reduced by TNFR2 knockout (*Figure 2—figure supplement 1B*) while DP cells, SP CD4$^+$ and SP CD8$^+$ cells were unchanged (*Figure 2—figure supplement 1C–E*). Transcription of *Cxcl10*, *Sting1,* and *TNF* was unaltered (*Figure 2—figure supplement 1F–H*). However, transcription of *Il1b* was elevated in TNFR1 and TNFR2 knockout mice (*Figure 2—figure supplement 1I*), albeit unaltered in TNFR1/2 knockout mice in comparison to STING WT;BL6. The total splenic cell content was not altered in TNFR1 and TNFR2 knockout mice, respectively (*Figure 2—figure supplement 1J*) and the numbers of CD4$^+$ and CD8$^+$ T cells were reduced in the spleens of TNFR1/2 knockout mice (*Figure 2—figure supplement 1K and L*). Numbers of mononuclear cells did not vary between these genotypes (*Figure 2—figure supplement 1M and N*). These observations are in accordance with previously published data (*Erickson et al., 1994*; *Peschon et al., 1998*; *Pfeffer et al., 1993*).

In STING ki mice lacking TNFR1 or TNFR1/2, we found that total thymic cellular count was slightly elevated (*Figure 2A*). The absolute numbers in the DN and DP stages were unaffected in STING ki mice by the lack of TNFR1 and/or TNFR2 (*Figure 2B and C*). However, deletion of TNFR1/2 signaling induced a significant increase in cellular count of thymic SP CD4$^+$ and SP CD8$^+$ in STING ki mice (*Figure 2D and E*).

Interestingly, disruption of TNFR signaling resulted in lower transcription of various signaling pathways. Thymocytes of STING ki mice lacking TNFR1/2 expressed significantly lower levels of IFN-related genes (*Cxcl10*, *Sting1*) (*Figure 2F and G*) and mice lacking TNFR1 and TNFR1/2 expressed reduced levels of NF-κB-related genes (*Tnf*, *Il1b*) compared to STING ki mice with functional TNFR (*Figure 2H and I*). Obviously, lack of TNFR1 signaling pathways resulted in marked reduction of proinflammatory transcription, potentially causing improvement of physiological function of the thymus in STING ki mice.

In addition, absence of both TNFRs together resulted in an attenuated severity of splenomegaly (*Figure 2J*). The functional loss of TNFR1 or TNFR2 alone had no impact on the manifestation of splenomegaly in STING ki mice. We observed a decrease in splenic CD4$^+$ T cell numbers in STING ki mice lacking TNFR2 (*Figure 2K*). Numbers of splenic CD4$^+$ and CD8$^+$ T cells were similar in STING ki;*Tnfr1$^{-/-}$* and STING ki;*Tnfr1/2$^{-/-}$* mice (*Figure 2K and L*). Furthermore, the spleens contained significantly fewer myeloid cells (monocytes and neutrophils) in STING ki;*Tnfr1/2$^{-/-}$* mice compared to STING ki;BL6 mice (*Figure 2M and N*). Taken together, constitutive activation of STING in STING ki mice severely affected SP CD4$^+$ and SP CD8$^+$ T cells in the thymus. The content of CD8$^+$ T cells in spleens of STING ki mice was independent of TNFR signaling. However, the numbers of CD4$^+$ T cells in STING ki;*Tnfr2$^{-/-}$* mice were dependent on TNFR2 signaling and the presence of myeloid cells in the spleen of STING ki mice depended on combined signaling of TNFR1 and TNFR2.

## Lack of TNFRs does not restore the formation of lymph nodes in STING ki mice

Constitutive activation of STING N153S in mice led to blockade of lymph node development (*Bennion et al., 2020*). After dye injection, we detected stained popliteal and iliac lymph nodes only in STING WT mice, not in STING ki mice. Deletion of TNFR1 or/and TNFR2 had no influence on lymph node development in STING ki mice (*Figure 2—figure supplement 2A and B*).

## Neuroinflammation and neurodegeneration in dependency of TNFR1/2 signaling

The extent of inflammation in mouse brain resulting from constitutive activation of STING N153S was reported by quantifying the density of Iba1-positive microglia (*Figure 3A*). Consistent with our previous findings (*Szego et al., 2022*), the density of Iba1-positive microglia in the *substantia nigra*

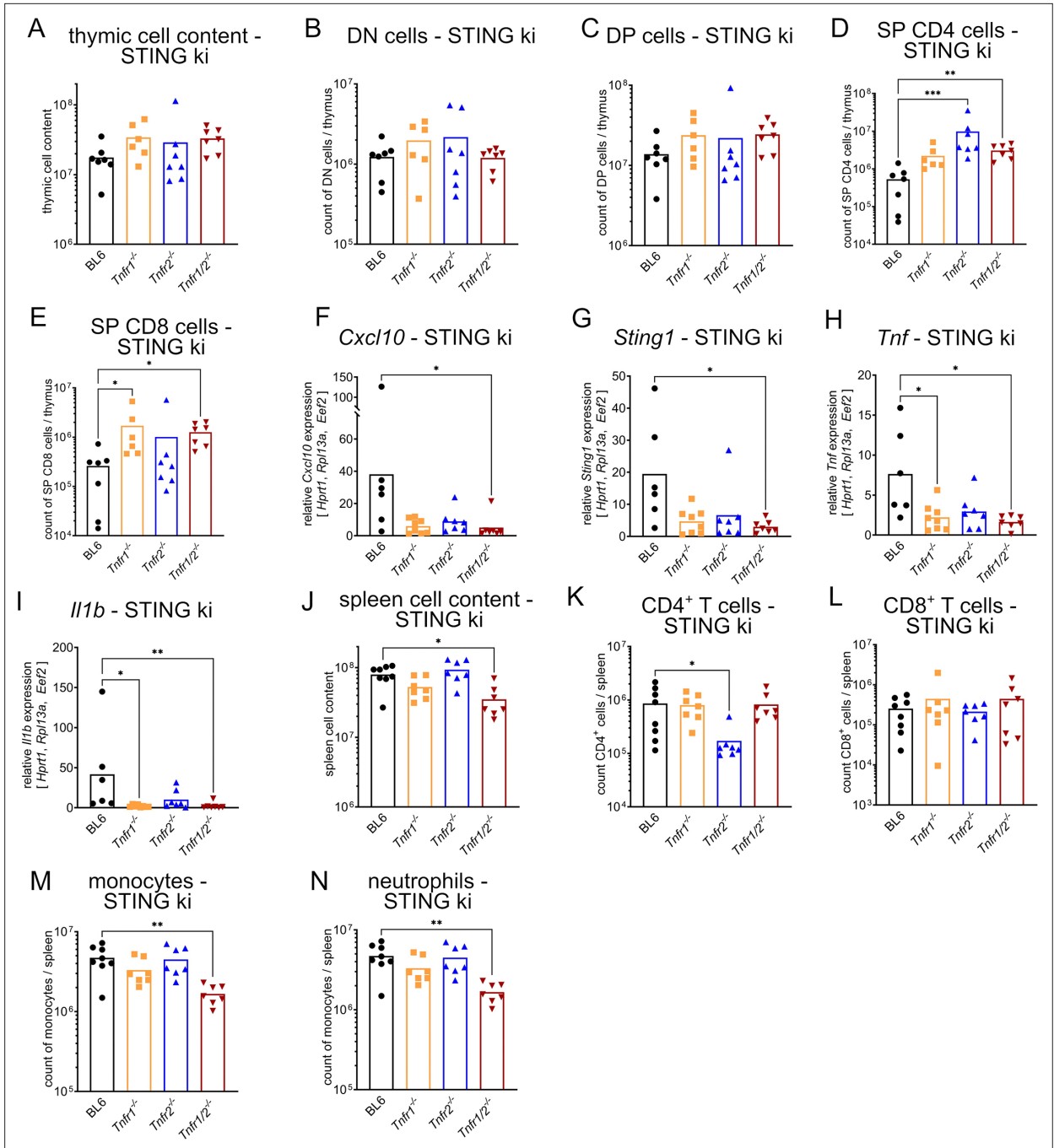

**Figure 2.** Inhibition of TNFR signaling regulates frequencies and numbers of thymic and splenic cells in STING ki mice. (**A**) Cellular count of all isolated cells per thymus in STING ki mice of indicated genotype. (**B**) Numbers of DN, (**C**) DP, (**D**) SP CD4+, and (**E**) SP CD8+ thymocytes per thymus in STING ki mice of indicated genotype. (**F**) Relative gene expression of *Cxcl10*, (**G**) *Sting1*, (**H**) *Tnf,* and (**I**) *Il1b* in thymus tissue from STING ki mice of indicated genotype. (**J**) Cellular count of all isolated cells per spleen in STING ki mice of indicated genotype. (**K**) Number of splenic CD4+ T cells, (**L**) splenic CD8+ T cells, (**M**) splenic monocytes, and (**N**) splenic neutrophils in STING ki mice of indicated genotypes. Markers represent individual mice, bars represent mean of n=7–8 mice per group pooled from nine independent preparations analyzed by Kruskal-Wallis test including Dunn's multiple comparisons test. *p<0.05, **p<0.005, ***p<0.001.

The online version of this article includes the following figure supplement(s) for figure 2:

**Figure supplement 1.** Inhibition of TNFR signaling did not affect frequencies and numbers of thymic and splenic cells in STING WT mice.

**Figure supplement 2.** Inhibition of TNFR signaling could not restore the development of lymph nodes.

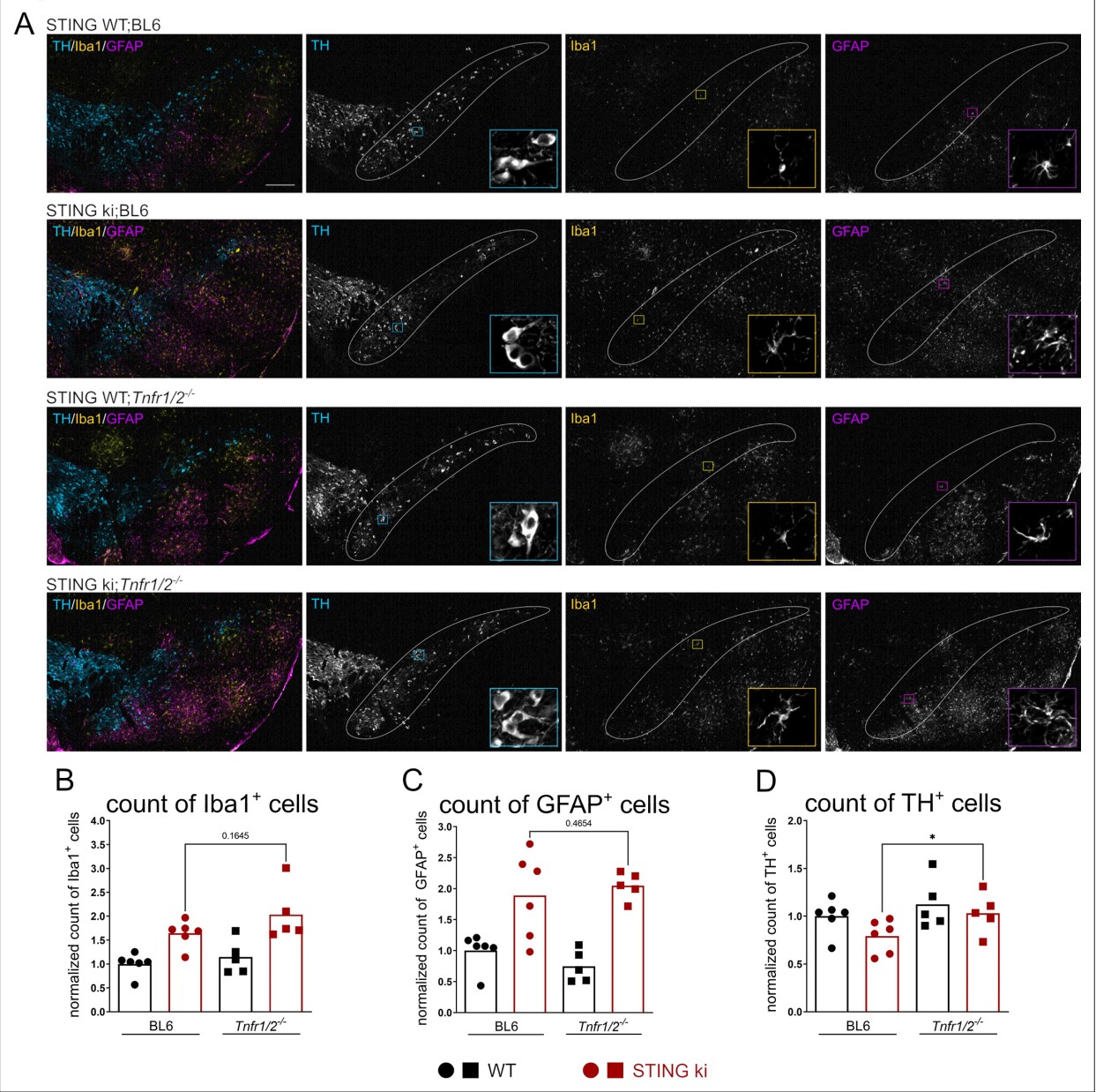

**Figure 3.** Lack of TNFR signaling improves the number of dopaminergic neurons in the *substantia nigra* of STING ki mice. (**A**) Representative images of TH-positive dopaminergic neurons, Iba1-positive microglia, and GFAP-positive astrocytes in the *substantia nigra pars compacta* (SNc, encircled area) of indicated genotypes. Scale bar represents 200 μm. (**B–D**) Number of Iba1-positive (**B**), GFAP-positive (**C**), TH-positive (**D**) cells in the SNc of the indicated genotypes expressed relative to the number of TH-positive neurons in the SNc of the corresponding mouse line without STING ki. Markers represent individual mice. Bars represent mean of all n=5–6 per group pooled from two independent preparations. Analysis by Mann-Whitney test. *p<0.05.

was higher in STING ki;BL6 mice than in STING WT mice (*Figure 3B*). TNFR deficiency did not affect neuroinflammation because there was no significant difference between the density of Iba1-positive microglia between STING ki;BL6 mice and STING ki;*Tnfr1/2⁻ᐟ⁻* mice (*Figure 3B*). This suggests that the TNF pathway is not required for STING-induced microglia activation in the *substantia nigra*.

In addition, we measured the extent of STING-induced astrogliosis by quantifying the density of GFAP-positive cells (*Figure 3A*). Consistent with our previous findings, the density of GFAP-positive astroglia was higher in STING ki than in STING WT mice (*Figure 3C*). Yet, as for microglia, there was no significant difference between the density of GFAP-positive astroglia between STING ki;BL6 mice

and STING ki;*Tnfr1/2⁻/⁻* mice (*Figure 3C*), suggesting that the TNF pathway is not required for STING-induced astrogliosis in the *substantia nigra*.

Finally, we measured the extent of STING-induced neurodegeneration by quantifying the density of TH-positive dopaminergic neurons in the *substantia nigra* (*Figure 3A*). As in our previous findings, the density of TH-positive neurons was lower in STING ki;BL6 mice than in STING WT mice (*Figure 3D*). The density of TH-positive neurons in the *substantia nigra* of STING ki;*Tnfr1/2⁻/⁻* mice was higher than the density of TH-positive neurons in the *substantia nigra* of STING ki;BL6 mice (*Figure 3D*), suggesting that the STING-induced degeneration of TH-positive neurons was blunted in *Tnfr1/2⁻/⁻* mice and that TNFR1/2 are involved in the STING-induced degeneration of dopaminergic neurons.

Hence, there is a discrepancy between STING-induced effects on glial cells as opposed to STING-induced effects on neurons. The dependence of STING-induced neurodegeneration but not glial response on TNFR1/2 suggests that the STING-induced degeneration of dopaminergic neurons is not a direct consequence of microglia or astroglia activation. This is consistent with the emerging concept of a neuron-specific inflammatory response (*Welikovitch et al., 2020*).

## TNFR1 signaling drives SAVI-associated lung inflammation in STING ki mice

STING N153S gain-of-function mutation induces lethal inflammatory lung disease, a hallmark of SAVI disease. We previously demonstrated that interstitial lung disease developed largely independent of the IFN I signaling, and occurs in the absence of cGAS, IRF3, IRF7, and of IFNAR1 (*Luksch et al., 2019*).

Inactivation of TNFR2 in STING ki mice only mildly reduced the transcription of SAVI-associated cytokines in lung tissue, while lack of TNFR1 led to a much stronger reduction of interferon-driven *Cxcl10*, *Sting1,* and NF-κB-driven *Tnf* or *Il1b* transcripts. Strikingly, co-deletion of both, *Tnfr1/Tnfr2* genes, completely rescued the inflammatory transcriptional signature in lungs of STING ki mice compared to STING WT mice (*Figure 4A–D*; *Figure 4—figure supplement 1A–D*). Interestingly, on the protein level, only loss of TNFR1 or TNFR1/2 signaling reduced the amounts of produced CCL2 and IL-6 in lungs of STING ki mice, while loss of TNFR2 signaling alone had no effect (*Figure 4E and F*; *Figure 4—figure supplement 1E and F*). However, this effect appeared to be tissue-specific, as we failed to detect a reduction of systemic proinflammatory cytokines in serum (*Figure 4—figure supplement 1H–O*).

In our STING ki mice, carrying the N153S mutation, approximately 14% of the lung area were infiltrated by immune cells (*Figure 4G and H*; *Figure 4—figure supplement 1G*). Development of interstitial lung disease in STING ki mice was almost completely prevented by inactivation of TNFR1 (infiltration<0.5% of lung area) but not of TNFR2. Likewise, STING ki;*Tnfr1/2⁻/⁻* mice were completely devoid of lung inflammation. Collectively, our data suggest that the secretion of inflammatory cytokines and subsequent inflammation of lungs in STING ki mice are driven by aberrant signaling through TNFR1.

## Lack of TNFR1/2 abrogates pathologic phenotype in primary lung endothelial cells

Lung inflammation in STING ki mice manifested around pulmonary blood vessels. We hypothesized that lung endothelial cells could be involved in the development of interstitial lung disease. To address this, we isolated primary murine lung endothelial cells from STING WT, STING ki, and STING WT;*Tnfr1/2⁻/⁻* and STING ki;*Tnfr1/2⁻/⁻* mice and subjected them to bulk RNAseq. Analysis of the primary lung endothelial transcriptomes revealed a decreased transcription of several proinflammatory cytokines (e.g. *Tnf*, *Il1b*) and chemokines (e.g. *Cxcl1*, *Cxcl2*, *Cxcl9*, *Cxcl10*, *Ccl2*, *Ccl3,* and *Ccl4*) in STING ki mice lacking TNFR1/2 compared to STING ki mice (*Figure 5A–C*). Interestingly, chemokines CCL2, CCL3, and CCL4 are essential for the attachment of leukocytes and subsequent migration across the endothelial barrier (*Roblek et al., 2019*; *Stamatovic et al., 2003*). Furthermore, we observed a strongly reduced expression of several cell adhesions molecules (*Jam3*, *Itgam*, *Vcam1*, *Glycam1*, *Madcam1*, *Ncam2,* and *Icam1*) and matrix metalloproteinase 9 (*Mmp9*), all of which are essential for transmigration of leukocytes. This suggests that loss of complete TNFR signaling reverts the inflammatory state of primary lung endothelial cells in STING ki mice, including their transcriptional transmigration signature.

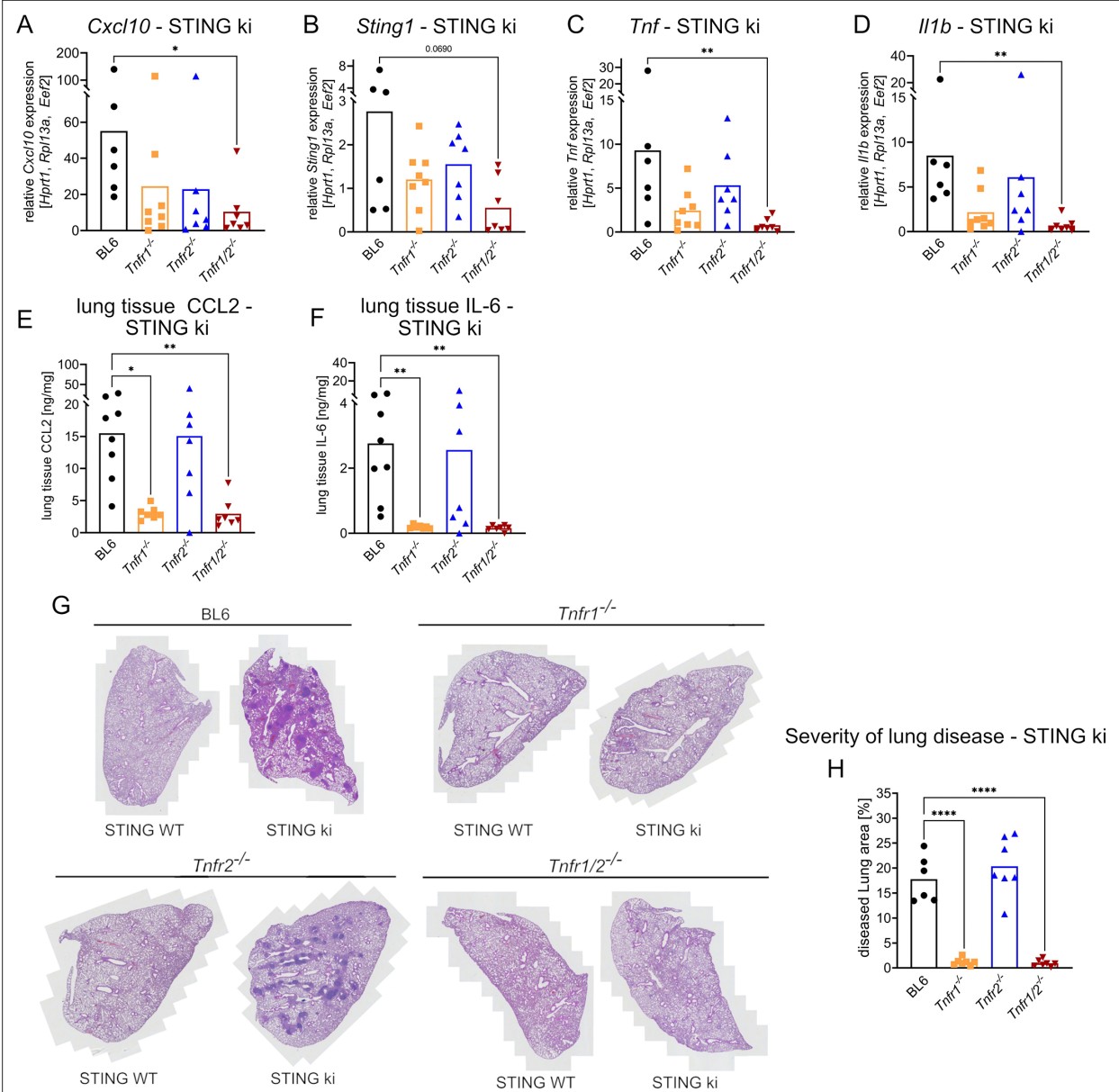

**Figure 4.** Knockout of TNFR signaling prevents manifestation of severe inflammatory lung disease in STING ki mice. (**A**) Gene expression of *Cxcl10*, (**B**) *Sting1*, (**C**) *Tnf*, and (**D**) *Il1b* in lung tissue from STING ki mice of indicated genotype. (**E**) Content of CCL2 and (**F**) IL-6 in lung tissue extracts from STING ki mice of indicated genotype. (**G**) Representative H/E lung sections of 10-week-old STING WT and STING ki mice of indicated genotype. (**H**) Quantification of lung disease severity from STING ki mice of indicated genotypes, data were analyzed by one-way ANOVA including Dunnett's multiple comparisons test. Markers represent individual mice, bars represent mean of n=7–8 mice per group pooled from nine independent preparations analyzed by Kruskal-Wallis test including Dunn's multiple comparisons test. *p<0.05, **p<0.005, ****p<0.0001.

The online version of this article includes the following figure supplement(s) for figure 4:

**Figure supplement 1.** Knockout of TNFR signaling did not affect the content of serum chemokines and cytokines.

Currently it is unclear if immune activation of lung endothelium is functionally involved in the development of SAVI lung disease. To address this, we established a cell culture system for quantification of neutrophil adhesion and neutrophil transmigration across a confluent endothelial cell monolayer under flow. Freshly isolated neutrophils from bone marrow of mice were added to cultured primary lung endothelial cell monolayers. All cells were exposed under constant flow pressure, which mimics physiological shear flow conditions. Quantification of attached and transmigrated neutrophils was performed by real-time microscopic supported video documentation. In the first setup, we used

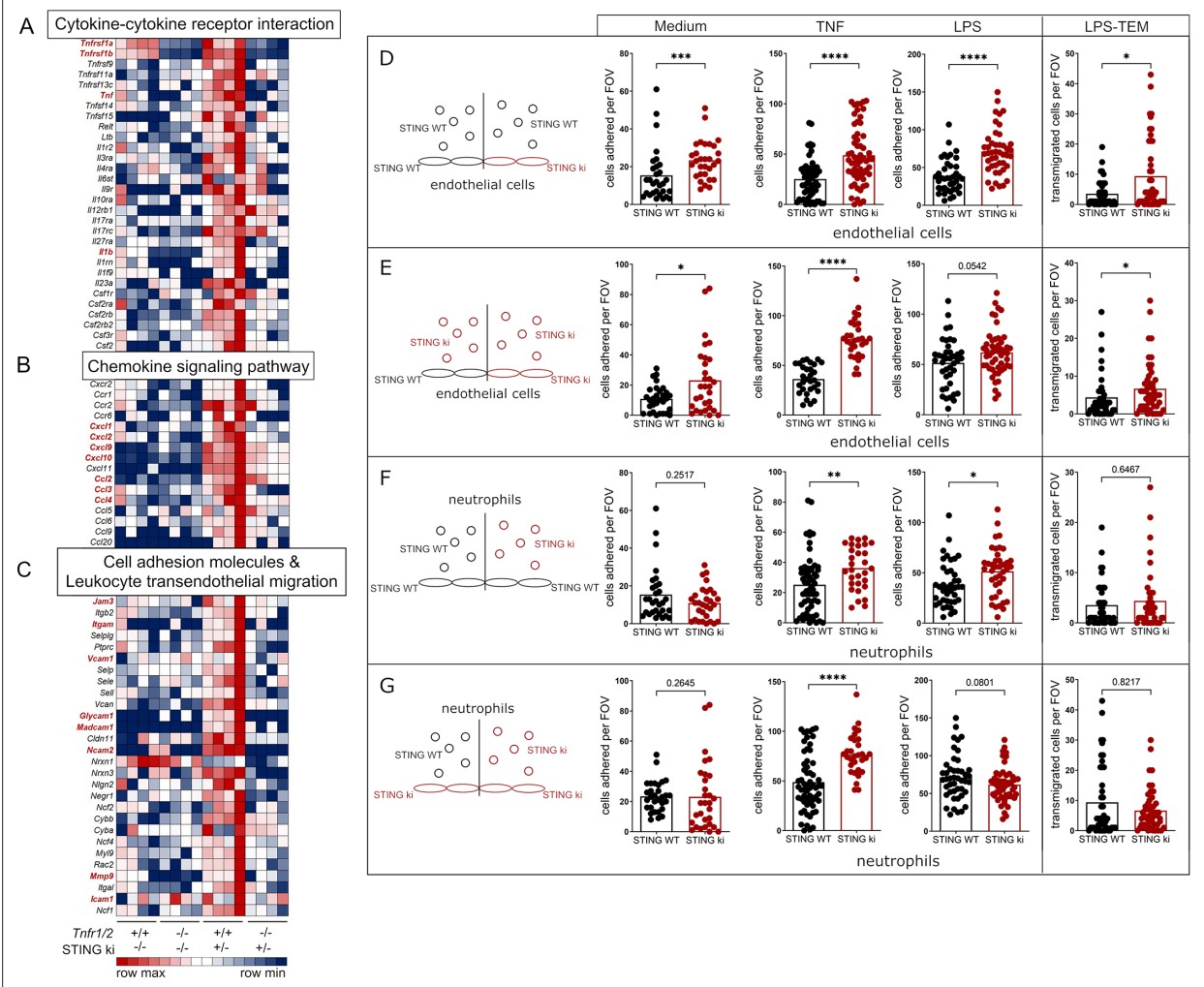

**Figure 5.** TNFR signaling is required for the transcriptional upregulation of inflammatory mediators and adhesion factors in murine lung endothelial cells from STING ki mice. Heatmap of normalized read counts for indicated transcript, summarized in specific pathways after bioinformatics analysis using DAVID (Database for Annotation, Visualization, and Integrated Discovery, LHRI). (**A**) Cytokine-cytokine receptor interaction. (**B**) Chemokine signaling pathway. (**C**) Cell adhesion molecules and leukocyte transendothelial migration. Remarkable genes are highlighted in bold and red letters. (**D–G**) Analysis of neutrophil attachment and transmigration across endothelial cell monolayers under flow. Schematic representations (left) of experimental setup, circles demonstrate neutrophils; ovals demonstrate endothelial cells, black shapes for STING WT, and red shapes for STING ki genotype. All experimental setups were performed with endothelial cell monolayer without preincubation (medium) or preincubation with TNF or LPS. Quantification of attached neutrophils (medium, TNF, LPS) and transendothelial migrated (TEM) neutrophils (after LPS preincubation = LPS-TEM). (**D**) Influence of STING ki endothelial cells compared to STING WT endothelial cells in attachment and transmigration of STING WT neutrophils. (**E**) Attachment and transmigration of STING ki neutrophils across the endothelial cell monolayer of indicated genotypes (STING WT or STING ki). (**F**) Influence of STING WT or STING ki neutrophils on their attachment and transmigration on STING WT endothelial cell monolayer. (**G**) Attachment and transmigration of STING WT or STING ki neutrophils across the STING ki endothelial cell monolayer. Markers represent separate measurements, bars represent mean of n=6–12 murine lung endothelial monolayers with five analyzed fields of view per sample analyzed by Mann-Whitney test. *p<0.05, **p<0.005, ***p<0.001, ****p<0.0001.

cultured endothelial cells from STING WT and STING ki mice, respectively (*Figure 5D and E*). Isolated neutrophil cells of STING WT mice attached significantly more frequently to STING ki endothelial cells than to STING WT endothelial cells (*Figure 5D*), even without preincubation of the endothelial cell monolayer. We previously demonstrated that chronic activation of STING in STING N153S mice induced elevated transcription and production of TNF in the lung tissue. To mimic this, we preincubated the endothelial cell monolayer with TNF overnight. The attachment of STING WT neutrophils was much stronger after preincubation of endothelial cells with TNF compared to untreated cells. The reinforcement of proinflammatory signaling after preincubation with TNF resulted in elevated counts

of attached (STING WT) neutrophils on STING ki endothelial cell monolayer compared to STING WT endothelial cell monolayers.

Many SAVI patients suffer from recurrent bacterial infections in the lungs (*Liu et al., 2014*). For analysis of endothelial cell function during bacterial infection, we preincubated the endothelial cell monolayer with LPS overnight. Similar to TNF pretreatment, the attachment of STING WT neutrophils was increased on LPS-exposed STING ki endothelial cells compared to STING WT. LPS also increased transendothelial migration (TEM) of STING WT neutrophils across the STING ki endothelial cell monolayer compared to STING WT endothelial cells (*Figure 5D*).

Next, we used neutrophils from STING ki mice for the investigation of STING WT and STING ki endothelial cell function (*Figure 5E*). More STING ki neutrophils attached to STING ki than to STING WT endothelial cell monolayer, independent of their preincubation. Similarly, we observed that significantly more neutrophils transmigrated across STING ki endothelial cell monolayer compared to STING WT endothelial cell monolayers. We conclude that attachment and transmigration of neutrophil cells were dependent on expression of STING gain-of-function mutation in endothelial cells. Taken together, STING ki endothelial cells supported the process of attachment and transmigration significantly more than STING WT endothelial cells.

In the next setup, we investigated the influence of the neutrophil genotype on the process of cell adhesion and transmigration (*Figure 5F and G*). Only after TNF or LPS preincubation of endothelial cell monolayer (STING WT), we observed an effective cell attachment. However, we could not detect any differences in transmigration of neutrophil cells of both genotypes (*Figure 5F*). This is in line with the observation of STING WT or STING ki neutrophil cell attachment on STING ki endothelial cell monolayer (*Figure 5G*). We did not detect any differences in cell attachment without pretreatment and in transmigration of STING WT and STING ki neutrophils. Taken together, STING ki endothelial cells promote neutrophil attachment and transmigration independent of neutrophil genotype (STING WT or STING ki). Attachment and transmigration of leukocytes are elementary mechanisms in the manifestation and progression of SAVI-driven inflammatory lung disease in STING ki mice.

Collectively, we demonstrate that lung inflammation of murine SAVI disease in STING N153S mice activated lung endothelial cells leading to increased attachment and transmigration of immune cells. Furthermore, our data suggests a pivotal role of TNFR1 signaling in the development of interstitial lung disease, which might have major implications for the treatment of human SAVI and other pulmonary inflammatory conditions with similar clinical symptoms.

## Discussion

In this work, we performed the inhibition of TNF signaling by generation of TNFR signaling-deficient mouse model with simultaneous chronic activation of STING caused by STING ki reduced some – but not all – consequences of constitutively active STING.

Constitutive activation of STING results in uncontrolled elevation of NF-κB signaling pathways, demonstrated by upregulated *Tnf* expression in STING ki mice (*Szego et al., 2022*).

For in-depth analysis of STING-dependent TNF expression, we used TNFR signaling-deficient STING ki mice. We observed improvement of thymocyte counts in STING ki mice lacking TNFR1 and TNFR1/2. Blockade of TNFR signaling, especially signaling of TNFR1, increased survival of thymocytes (SP CD8 cells) and enabled the enlargement of the peripheral T cell pool. Interestingly, the frequency of naïve T cells and effector T cells was unchanged in all analyzed STING ki mice. This indicates that blockade of TNFR signaling can promote the cellularity of thymocytes, but does not influence the activation of peripheral T cells. These results are in agreement with previously published data showing that systemic inflammation driven by TNFR signaling induced severe thymic atrophy. This phenotype was rescued totally in a TNFR1-deficient background (*Belhacéne et al., 2012*). We previously reported that STING ki mice show a disturbed development of lymph nodes leading to the complete loss of lymph nodes (*Bennion et al., 2020*). We found that the absence of TNFR1 signaling did normalize the thymocyte numbers, but did not restore the development of lymph nodes in STING ki mice. In contrast, STING ki mice lacking the IFNGR1 showed a successful lymph node development, but no improvement of thymocyte development (*Stinson et al., 2022*). These observations suggest that T cell development in the thymus is dependent on TNFR1 signaling, but the development of lymph nodes depends on IFNGR1 signaling.

Manifestation and progression of severe interstitial lung disease is a dominant hallmark of the murine systemic autoinflammatory SAVI disease (*Warner et al., 2017*). It was not possible to improve the severity of lung disease by a curative bone marrow transplantation in STING ki mice (*Luksch et al., 2019*). We observed a strong increase of type I IFN and type II IFN signaling as well as elevated transcription of proinflammatory mediators, e.g., *Tnf* and *Il1b* in lung tissue of STING ki mice. The manifestation of this inflammatory lung disease was independent of type I IFN, but depended on type II IFN signaling (*Stinson et al., 2022*). Our results demonstrated impressive that the initiation and progression of severe lung disease in STING ki mice is also dependent on TNFR1 signaling. In contrast, the deletion of TNFR2 in STING ki mice did not improve the severity of inflammatory lung disease. This is in line with a report that TNFR1 deficiency inhibited skin inflammation and TNFR2 deficiency rather promoted the development of skin disease in a mouse psoriasis model (*Chen et al., 2021*).

The STING-induced degeneration of dopaminergic neurons in the substantia nigra was reduced in mice with TNFR1/2 deficiency (*Figure 3D*), whereas the activation of astroglia and microglia was not (*Figure 3B and C*). Neurons express both TNFRs, so circulating TNF could affect dopaminergic neurons without involvement of glial cells. Indeed, elevated levels of TNF have been implicated in the degeneration of dopaminergic neurons (*Harms et al., 2021*; *Williams-Gray et al., 2016*). For instance, anti-TNF therapy reduced the incidence of PD in patients with inflammatory bowel disease (*Peter et al., 2018*), and polymorphisms in the TNF gene have been associated with an increased risk for PD (*Chu et al., 2012*; *Nishimura et al., 2001*).

The preserved activation of glial cells in STING ki;*Tnfr1/2*$^{-/-}$ mice indicates that their activation does not require TNFRs. Genetic inactivation of *Casp1* in STING ki mice blocked IL-1β activation, NRLP3 inflammasome formation, and activation of astroglia but not microglia (*Szego et al., 2022*), suggesting that astroglia activation could be mediated by NLRP3-IL-1β signaling. Activation of microglia was affected only moderately by genetic inactivation of interferon receptor 1 (*Ifnar1*$^{-/-}$), suggesting that it is not dependent on this pathway.

For determination of essential mediators in the manifestation of SAVI characteristic lung disease, we performed a transcriptome analysis of freshly isolated murine lung endothelial cells. We found a significantly decreased expression of various proinflammatory mediators and their receptors in the lung endothelial cells of STING ki;*Tnfr1/2*$^{-/-}$ mice in comparison to STING ki mice. Remarkably, transcription of type II IFN-driven *Cxcl9*, *Cxcl10,* and NF-κB-driven *Tnf* and *Il1b* was downregulated in all analyzed STING ki mice with deletion of TNFR1 and TNFR2 compared to STING ki mice. This is in line with previously described results that lack of IFNGR1 in STING ki mice prevent uncontrolled upregulation of *Cxcl9* and *Cxcl10* gene expression (*Stinson et al., 2022*). The complete evaluation of RNA sequencing data of primary lung endothelial cells disclosed the involvement of chemokines and adhesions proteins in the manifestation of SAVI characteristic inflammatory lung disease. Transcription of *Ccl2*, *Ccl3,* and *Ccl4* was downregulated in endothelial cell of STING ki mice lacking TNFR1/2 signaling. The chemokine CCL2 induces cytoskeletal alterations in endothelial cells and is essential for recruitment and migration of neutrophil granulocytes across the endothelial barrier in mouse tumor models (*Roblek et al., 2019*). Additionally, endothelial secretion of CCL2 controls metastasis by promoting tumor cell extravasation (*Wolf et al., 2012*). In mice, CCL3 recruits neutrophil granulocytes to the lung in response to IFNγ-mediated signaling in a virus infection model (*Bonville et al., 2009*). In LPS-induced lung inflammation, CCL2 recruits macrophages and neutrophil granulocytes into lung tissue across the endothelial barrier (*Mercer et al., 2014*). The presence of CCL2 induces brain endothelial hyperpermeability and attracts leukocytes to murine brain endothelial cells (*Stamatovic et al., 2003*). Human gain-of-function mutations of *STING1* induced elevated transcription of *CCL3*, *CCL4*, and *IL6* in PBMCs from SAVI patients (*de Cevins et al., 2023*). Taken together, the increased expression of endothelial *Ccl2*, *Ccl3,* and *Ccl4* is necessary for leukocyte transmigration into the lung tissue in the context of virus- or bacteria-induced inflammation, autoinflammation, and tumor-promoting metastasis. For transmigration of cells across the endothelial barrier, the presence of these chemokines is essential (*Mercer et al., 2014*).

Constitutively active STING induces activation of different pathways, e.g., interferon- or NF-κB-driven signaling. Our results demonstrate that proinflammatory mediators are much more produced in primary lung endothelial cells of SAVI mice. We assume that this uncontrolled signaling is essential for endothelial barrier dysfunction and finally for accumulation of leukocytes in the lung tissue. Primary lung endothelial cells of STING ki mice allowed more attachment of neutrophils compared to primary

lung endothelial cells of STING WT mice. We pointed out that attachment and transmigration under native condition was independent of neutrophil genotype. These results demonstrated clearly that the intact barrier of lung endothelial cells is a critical factor for the manifestation of severe inflammatory lung disease. In a murine peritonitis model, it was demonstrated that the expression of STING in endothelial cells is essential for leukocyte transmigration (*Anastasiou et al., 2021*). Endothelial cells isolated from human coronary artery produce high amounts of adhesions proteins, e.g., ICAM1, that support the transendothelial migration of leukocytes (*Xue et al., 2018*). In human umbilical vein endothelial cells (stimulated with TNF), several alterations in gene transcription were observed, e.g., elevated transcription of proinflammatory mediators and adhesion proteins as well as decreased transcription of cytoskeletal components (*Rhead et al., 2020*; *Zhou et al., 2002*). This altered signaling led to molecular alterations in endothelial cells and disrupted the endothelial cell barrier. Interestingly, the mouse model for acute lung injury induced by LPS inhalation is characterized by elevated STING expression (*Wu et al., 2022*). Pharmacological inhibition of STING activation prevented the manifestation of this disease. In the same line it was published that the progression of ANCA-associated vasculitis is dependent on the activation of cGAS/STING/IRF3 axis (*Kessler et al., 2022*). Blockade of STING activation improved the severity of this disease significantly. Previous murine studies indicated that the expression of STING V154M in endothelial cells is only essential for manifestation of inflammatory infiltrates (*Gao et al., 2024*). Taken together, the chronic activity of STING in endothelial cells of STING ki mice is important for disruption of endothelial cell barrier and manifestation of severe lung disease.

Stinson and coauthors described that murine SAVI disease is promoted by IFNγ signaling (*Stinson et al., 2022*) and we here observed a TNFR1 signaling dependency of this disease. These observations demonstrated that the manifestation and progression of systemic murine SAVI disease is dependent on various signaling pathways.

Murine SAVI is caused by heterozygous point mutations in *Sting1*, resulting in constitutive activation of STING with uncontrolled inflammatory activity. T cell lymphopenia and interstitial lung disease are characteristics of murine SAVI disease comparable to symptoms of severe COVID-19 disease. Cell fusion caused by SARS-CoV-2 spike protein is a potent activator of cGAS-STING pathway with induction of type I IFN and cytokine production (*Berthelot et al., 2020*; *Liu et al., 2022b*). During SARS-CoV-2 infection, the secretion of TNF and IFNγ induces inflammatory cell death by PANoptosis mediated by JAK/STAT1/IRF1 axis (*Karki et al., 2021*). Infliximab treatment of patients with severe COVID-19 disease improved the numbers of blood CD4[+] T cells (*Popescu et al., 2022*). The TNFR signaling is an essential part in the progression of COVID-19 disease as well as murine SAVI disease. Inhibition of TNFR activity is beneficial for both diseases.

In summary, our work suggests that TNFR1 signaling is a driver of murine SAVI disease. Loss of TNFR1 signaling can restore thymocyte (SP CD8 cells) numbers. Lack of TNFR1 signaling prevented the severe inflammatory lung disease manifestation in STING ki mice. However, it is important to note that with these newly generated mouse lines of TNFR signaling blockade, we are not able to explain all features of this systemic autoinflammatory disease. Additional investigations are required for complete elucidation of involved mechanisms and for development of new therapeutic options for SAVI patients.

## Acknowledgements

We thank Katrin Höhne and Barbara Utess for excellent technical support. A Rösen-Wolff, LL Teichmann, C Günther, and R Behrendt were supported by the German Research Foundation (DFG) Project ID 369799452-TRR237. R Behrendt is additionally funded by the Deutsche Forschungsgemeinschaft (DFG, German Research Foundation) under Germany's Excellence Strategy – EXC2151-390873048.

## Additional information

### Funding

| Funder | Grant reference number | Author |
|---|---|---|
| Deutsche Forschungsgemeinschaft | 369799452-TRR237 | Felix Schulze |
| Deutsche Forschungsgemeinschaft | EXC2151 - 390873048 | Rayk Behrendt |
| Deutsche Forschungsgemeinschaft | 446167311 | Angela Rösen-Wolff |

The funders had no role in study design, data collection and interpretation, or the decision to submit the work for publication.

### Author contributions

Hella Luksch, Conceptualization, Data curation, Supervision, Investigation, Methodology, Writing – original draft, Project administration, Writing – review and editing; Felix Schulze, Data curation, Investigation, Methodology, Writing – review and editing; David Geißler-Lösch, David Sprott, Formal analysis, Investigation, Writing – original draft; Lennart Höfs, Investigation; Eva M Szegö, Rayk Behrendt, Data curation, Formal analysis, Investigation, Methodology, Writing – original draft; Wulf Tonnus, Formal analysis, Methodology, Writing – original draft; Stefan Winkler, Claudia Günther, Andreas Linkermann, Investigation, Methodology, Writing – original draft; Lino L Teichmann, Formal analysis, Funding acquisition, Investigation, Writing – original draft; Björn H Falkenburger, Data curation, Visualization, Methodology, Writing – original draft, Writing – review and editing; Angela Rösen-Wolff, Conceptualization, Resources, Funding acquisition, Writing – original draft, Writing – review and editing

### Author ORCIDs

Hella Luksch https://orcid.org/0000-0001-7070-4992
Felix Schulze https://orcid.org/0000-0002-8220-5012
David Geißler-Lösch https://orcid.org/0009-0006-8332-2145
Wulf Tonnus https://orcid.org/0000-0002-9728-1413
Claudia Günther https://orcid.org/0000-0002-4330-1861
Andreas Linkermann https://orcid.org/0000-0001-6287-9725
Rayk Behrendt https://orcid.org/0000-0002-1091-2877
Lino L Teichmann https://orcid.org/0000-0001-9489-7282
Björn H Falkenburger https://orcid.org/0000-0002-2387-526X
Angela Rösen-Wolff https://orcid.org/0000-0002-9613-5879

### Ethics

All mice experiments were approved by the Landesdirektion Sachsen (TVV 4/2019, TVV 13/2019) and carried out in accordance with the institutional guidelines on animal welfare.

### Decision letter and Author response

Decision letter https://doi.org/10.7554/eLife.101350.sa1
Author response https://doi.org/10.7554/eLife.101350.sa2

## Additional files

### Supplementary files

Supplementary file 1. List of antibodies for FACS analysis (all from BioLegend). The following antibodies were used for FACS analysis.

Supplementary file 2. List of antibodies for immunofluorescence staining of brain sections. The following antibodies were used for staining of brain sections.

Supplementary file 3. List of qRT-PCR primers. The following primers were used for qRT-PCR analysis.

Supplementary file 4. List of effect size and power calculations. Analysis of effect size and power

were performed by G*Power 3.1.9.4, effect size d convention (<0.5 = small, 0.5–0.8=medium, >0.8 = large effect), and effect size f convention (<0.25 = small, 0.25–0.4=medium, >0.4 = large effect). MDAR checklist

## Data availability

Transcriptomic data are deposited on GEO database, accession no GSE244062.

The following dataset was generated:

| Author(s) | Year | Dataset title | Dataset URL | Database and Identifier |
| --- | --- | --- | --- | --- |
| Behrendt R, Luksch H, Rösen-Wolff A | 2024 | Tissue-specific inflammation induced by constitutively active STING is mediated by enhanced TNF signaling | http://www.ncbi.nlm.nih.gov/geo/query/acc.cgi?acc=GSE244062 | NCBI Gene Expression Omnibus, GSE244062 |

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
