## [Editor Report]

This study provides important insights into the pathogenesis of systemic autoinflammatory disease STING-associated vasculopathy with onset in infancy (SAVI). Specifically, the authors demonstrate the critical role of tumor necrosis factor (TNF) signaling in SAVI in mouse model. Overall, the data presented are convincing, supporting the author's conclusions.

---

## [Decision Letter]

[Editors' note: this paper was reviewed by Review Commons.]

Thank you for submitting your article "Tissue inflammation induced by constitutively active STING is mediated by enhanced TNF signaling" for consideration by *eLife*. Your article has been reviewed by 3 peer reviewers at Review Commons, and the evaluation at *eLife* has been overseen by a Reviewing Editor and Tadatsugu Taniguchi as the Senior Editor.

Based on the previous reviews and the revisions, the manuscript has been improved but there are some remaining issues that need to be addressed, as outlined below:

We have read briefly the manuscript and accompanying comments, some of which are fair and some are not. We think this is a reasonable effort to understand the role of TNFRs in the pathogenesis of STING-associated vasculopathy. A significant amount of the disease observed is secondary to TNFRI activation and partially on TNFRII. Our greatest concern here relates to the use of Infliximab (an anti-human TNF antibody) rather than etanercept. There are studies which question the specificity of infliximab in the murine system (see for example, Anti‐inflammatory effects of infliximab in mice are independent of tumour necrosis factor α neutralization. Clin Exp Immunol. 2017 Feb; 187(2): 225-233. doi: 10.1111/cei.12872). Etanercept is known to bind to and inactivate both human and murine TNF. Although Infliximab has been used in the literature, its effects have been questioned over time and most recent studies preferentially use etanercept as a tool to block TNF-a in mice. The authors do present some results using etanercept (supplementary material). Unless the authors can show convincingly that murine TNF-a is being blocked in their system by Infliximab, this whole part of the study needs to be removed.

---

## [Author Response]

Reviewer #1:STING is a key signalling hub in the innate immune system, receiving multiple inputs from upstream activators (such as cGAS) and in turn triggering multiple downstream events (such as IFN induction, NF-κB signalling, autophagy, cell death). Mutations in the STING gene cause a rare inflammatory disease called SAVI. Using a previously established STING ki mouse that recapitulates some of the clinical observations in SAVI patients, this manuscript tests the hypothesis that TNF signalling drives pathology. Using anti-TNF antibody and TNF receptor knockout, the authors show that TNF indeed plays important roles in causing disease in this mouse model. For example, the loss of T cells and neurons is prevented when TNF signalling is blocked, and lung pathology is rescued in STING ki mice lacking TNF receptors. Overall, the manuscript is well written and laid out, and the experimental work is of a high technical standard.Major comments1. Most figures show pooled data from two independent experiments including a total of 5-8 mice. Given the variability in some of the readouts, this raises the question of whether there is sufficient statistical power to draw conclusions. For example, in Figure 2, the conclusion that "Infliximab did not alter the expression of inflammatory mediators" seems questionable given the results in Figure 2F and G. Did the authors perform a power calculation? What effect size can the authors detect given the variability and number of replicates? Similarly, in Figure 3, the authors conclude that "Disruption of TNFR signaling did not significantly prevent T cell lymphopenia"; however, with some more replicates, the data in Figure 3D would likely reach significance. Similar concerns apply to several panels in Figures 4 and 6 and to Figure S5M. Ideally, the authors should perform additional repeat experiments to increase the number of replicates. If that is not possible, power calculations need to be provided and conclusions should explicitly mention the minimum effect size that the author can detect given the small sample size (for example "Infliximab did not alter the expression of inflammatory mediators more than x-fold").

We thank the reviewer for his/her time and for the constructive comments. Below please find our detailed responses to your points.

Thank you for this suggestion. However, it is not possible to repeat the treatment of mice with Infliximab for generation of more replicates. The blockade of TNF signalling by treatment with drugs did not cure the murine SAVI disease. According to animal welfare restrictions, we cannot perform additional treatment experiments with Infliximab or Etanercept.

We analysed the effect size d, f and power of all these presented results and collected them in table S4. Additional explanations about effect sizes were added in the corresponding text to Figures 2 and 3. The demonstrated results in Figure 4 and 6 already contain significant data. We did not include the calculation of effects sizes here. All effect size and power calculations are summarized in table S4.

2. The authors should not make unjustified overstatements. For example, STING KI; TNFR1/2 KO mice should not be referred to as a "new mouse model". The manuscript simply tests the role of TNFR1/2 in the already published STING N153S model. In line 687, avoid using "impressively" and in line 734 avoid using "massively".

Thank you for this suggestion. We changed this sentence into:…”these newly generated mouse lines of TNFR”…., see line 796. Additionally, in line 687 (actual line 705) we omitted “impressively” and in line 734 “massively produced” into “elevated” (actual line 752).

Minor comments3. Line 767-769: The statement that spike activates cGAS is misleading, because this effect is an indirect consequence of cell-to-cell fusion (Liu et al. 2022).

Thank you for this suggestion. We changed this sentence into: Cell fusion caused by the SARS-CoV-2 spike protein is a potent… (actual line 785).

Reviewer #1 (Significance (Required)):The main strengths of this study are (1) the use of complementary antibody-based and genetic methods to test the role of TNF signalling; (2) the use of multiple different readouts; and (3) the analysis of many different cell types / organ systems. The main weaknesses are (1) small sample sizes limiting statistical power (see above) and (2) the exclusive use of mouse models.Overall, my opinion is that the advance is important, both fundamentally and clinically. Studies of this and the related V154M mouse model previously showed an important role of non-IFN pathways in driving disease. This study indicates that TNF signalling may cause pathology. This not only extends our understanding of STING's role in autoinflammation but also opens a direct therapeutic avenue using approved TNF targeting drugs.This study will be primarily of interest to specialised audiences working on STING and SAVI, and secondarily to the wider innate immunity field.This reviewer has expertise in the field of nucleic acid sensing, including cGAS-STING.

Reviewer #2:In this paper, Luksch et al. (2024) examines the role of TNF signaling in STING-associated vasculopathy with onset in infancy (SAVI). By using pharmacological inhibition and genetic inactivation of TNF receptors in a murine SAVI model (STING ki), the research found that pharmacologically inhibiting TNF signaling improved T cell lymphopenia but had limited effects on lung disease. Genetic inactivation of TNFR signaling, particularly TNFR1, enhanced thymocyte survival and expanded the peripheral T cell pool, reducing inflammation and neurodegeneration. The development and progression of severe lung disease in STING ki mice are also reliant on TNFR1 signaling, while TNFR2 deletion did not alleviate lung inflammation. The authors also explored the severe inflammatory lung disease manifestation, showing that primary lung endothelial cells in STING ki mice allowed more neutrophil attachment compared to those in STING WT mice, indicating chronic STING activity in endothelial cells disrupts the endothelial barrier and promotes severe lung disease. The study highlights TNFR signaling as crucial in SAVI and COVID-19 progression and suggests blocking TNFR1 signaling as a potential therapeutic approach for both diseases.Major comments:The paper establishes a strong connection between TNFR1 depletion and the reduction of SAVI disease severity in lung and neuroinflammation, suggesting TNFR1 blockade as a viable therapeutic strategy for SAVI. To strengthen the arguments and improve the therapeutic potential, the authors should address the following major comments:1. The authors conclude that TNFR1 signaling drives murine SAVI disease, as evidenced by the reduced severity of lung disease in TNFR1 -/- mice. While the genetic model is convincing, the discrepancy between pharmacological inhibition and genetic models needs clarification. Before attributing the pharmacological failure to late administration, have the authors considered that Infliximab might not sufficiently deplete TNF to achieve therapeutic benefits? In figure 2H, serum TNF levels were not significantly altered in STING ki mice treated with Infliximab. Have the authors considered using other TNF inhibitors or alternative methods to measure TNF depletion efficacy in STING ki murine models, such as qPCR, flow cytometry, or immunohistochemistry in lymph nodes or lung tissues?

We thank the reviewer for his/her time and for the constructive comments. Below please find our detailed responses to your points.

Thank you for this suggestion. In a preliminary experiment, we already treated STING WT and STING ki mice with Etanercept which is not included in the paper. 3-week-old mice were treated with subcutaneously injection of 25 mg/kg Etanercept or saline, twice per week, for 7 weeks. After treatment, all mice were euthanized and single cell suspensions of blood and spleen were used for flow cytometry analysis. Lung tissue was harvested for histological analysis. Quantification of gene expression was performed by snap frozen lung and kidney tissue and quantification of secreted proteins was analysed by snap frozen serum.

The transcription of ISGs and proinflammatory mediators in lung tissue was not significantly improved by the Etanercept treatment of mice, see Author response image 1 – D. Interestingly, the amount of secreted CXCL9 in the serum was reduced in Etanercept treated mice compared to vehicle treated mice (E). We concluded that our treatment strategy had no impact in the manifestation and progression of murine SAVI disease, in highly inflamed tissues / organs. However, we found a reduction (partially significant) of proinflammatory mediator transcriptions in the kidney of Etanercept treated mice compared to vehicle control mice. Murine SAVI disease is a systemic autoinflammatory disease without histological alteration in kidney tissue of 10 weeks old mice. Remarkably, transcription of ISGs and proinflammatory mediators is highly upregulated in SAVI mice. Treatment with Etanercept improved this aberrant gene expression in murine SAVI influenced tissue / organ (Author response image 1 – K). These results encouraged us to perform the treatment with infliximab because we expected a more pronounced effect since infliximab can bind the monomeric and trimeric form while etanercept can only bind to the active trimeric from of TNF.

**Author response image 1. sa2fig1:** Etanercept treatment of STING WT (in black) and STING ki (in red) mice. (**A**) Relative expression level of Cxcl10, (**B**) Mx1, (**C**) Tnf and (**D**) Il1b in lung tissue of Etanercept or saline treated STING WT and STING ki mice. (**E**) Quantification of CXCL9, (**F**) CXCL10, (**G**) IL-6 and (**H**) TNF in serum samples from STING WT and STING ki mice after treatment. (**I**) Relative expression level of Cxcl10, (**J**) Mx1, (**K**) Tnf and (**L**) Il1b in kidney tissue of treated mice.

2. The TNF pathway exhibits redundancy, as multiple signaling molecules or pathways can compensate for the loss of TNF function to maintain cellular processes and immune responses. The authors showed that thymocytes of STING ki mice lacking TNFR1/2 expressed significantly lower levels of IFN-related genes (Cxcl10, Sting1), and mice lacking TNFR1 and TNFR1/2 expressed reduced levels of NF-κB-related genes. Does this imply that IFN and NF-κB pathways are downstream of TNF signaling driving SAVI progression? It would be valuable to hear the authors' comments or postulations on the potential mechanisms of TNF driving SAVI progression in the discussion, and the methods to dissect the mechanisms further using genetic or pharmacological methods.

Thank you for this suggestion. STING is a key player in various proinflammatory mechanism and is directly involved in IFN and NF-κB signalling. We assume that these signalling pathways are adaptable to various proinflammatory situations. Knock out of TNFR1 and TNFR1/2 results in a strong inhibition of all inflammatory reactions in the whole organisms. We think, it is not possible to conclude mechanisms of murine SAVI manifestation and progression from the results of these mouse lines only. These observations provide new hypothesis, but cannot completely explain the mechanism.

3. The authors mentioned that the pharmacological inhibition of TNF by Infliximab is ineffective due to late administration compared to the onset of SAVI. How would this affect the therapeutic treatment of TNF if the treatment is going to be later than the disease onset? Can the authors elaborate on the potential ways to circumvent the timing of treatment? Would TNFR1 antagonists experience the same issue? To understand disease progression and optimal targeting times, the creation of an inducible TNFR1/2 -/- mouse model could be beneficial. This is optional, but the authors are encouraged to comment on improving TNFR1/2 -/- mouse SAVI models to further study the therapeutic potential of TNF signaling blockage in treating SAVI.

We agree with the suggestion. In the next project, we want to generate STING ki mice with inducible knock out.

Minor comments:4. The authors separate STING WT and STING ki into different graphs, which can sometimes make it hard to compare STING WT and STING ki baseline levels. It would be beneficial to merge the two genotypes into single graphs for easier comparison.

Thank you for this suggestion. In the first version of this manuscript, we collected results from STING WT and STING ki mice in one graph with 8 bars in different colours and textures in the case of TNFR knock out lines. These graphs were overloaded and very confusing. It is was not possible to mark statistical calculations inside these graphs without losing the focus. Hence, we created the demonstrated design of graphs. We think this is the most convincing version.

5. Figure S5 lacks statistical annotations, although the legends mention them. Are the statistics usually shown when a comparison is mentioned in the text, or are they only displayed when the differences are significant? It would be helpful if the authors could clarify this and ensure that all relevant statistical comparisons are clearly reflected in the graphs, regardless of the significance level. This consistency would improve the clarity and interpretation of the data presented.

Thank you for this suggestion. We removed the significance level from the legend of Figure S5 (actually line 1199).

6. The authors did an excellent job discussing the study's implications, but some of this content could be moved to the introduction. The hypothesis that "tumor necrosis factor (TNF) signaling is involved in the manifestation and progression of murine SAVI disease" can be introduced more naturally once the authors present previous findings on TNF's association with various autoimmune disorders. This would set a clear context for the study's objectives and rationale.

We agree with this suggestion and inserted the sentence: “In our previous investigations, we observed an elevated transcription of *Tnf* in spleen and thymus of STING ki mice (Siedel et al., 2020).” (actual line 89/90).

General Assessment: The study identifies enhanced TNF signaling as a driver of SAVI and specifies TNFR1 blockage as a promising treatment to reduce disease severity. It thoroughly characterizes pharmacological inhibition and genetic perturbations of TNF signaling in murine SAVI models and creates a novel mouse model for studying TNF-targeted therapies in SAVI treatment.However, the study is limited in characterizing the discrepancy between pharmacological inhibition and genetic depletion of TNF and understanding the underlying mechanisms of TNF driving chronic STING activation and tissue inflammation.Advances: The study extends knowledge in the field by demonstrating that enhanced TNF signaling drives SAVI, establishing causation rather than mere correlation. The authors provide strong rationale for treating SAVI with TNF inhibitors/blockage, previously used in other autoimmune disorders like IBD or Crohn's disease, but not in SAVI. They also present a valuable genetic model for studying TNFR signaling blockage in SAVI progression, which is important for both the field of SAVI and future therapy development.Audience: The research provides translational and clinical insights by suggesting that targeting TNFR1 signaling could inspire novel treatments for SAVI. The study also advances basic research on SAVI disease progression. Immunologists and clinicians studying and treating autoimmune disorders are the intended audience, but the findings have broader implications. The study highlights the potential role of TNF signaling in COVID-19 disease progression and treatment, thus attracting interest beyond the field of autoimmune disorders.Field of expertise:cGAS-STING regulation in chromosomally unstable cancers, genomic instability, nuclear envelope rupture and repairDo not have sufficient expertise in:Immunological underpinning of autoimmune disorders, clinical models or manifestations of SAVI

Reviewer #3:Uncontrolled activation of STING is linked to autoinflammatory disease "STING-associated vasculopathy with onset in infancy (SAVI)". The authors had previously published a mouse model of SAVI, which was generated by knocking in the disease causing variant N153S into the endogenous murine Sting1 gene (STING ki) (Luksch et.al., 2019). In the current study, the author further investigated the role of tumor necrosis factor (TNF) signaling in manifestation and progression of murine SAVI disease by using the approach of pharmacologic and genetic inhibition of TNF receptors TNFR1 and TNFR2. Overall, the authors were able to demonstrate the following novel findings:1) Infliximab treatment of STING ki mice significantly increased the number of blood CD8^+^ T cells and thymic cells count. The authors claimed that the pharmacological inhibition of TNF signalling has a partial rescue effect of T cell lymphopenia. However, pharmacologic inhibition of TNF signalling however has no effect on lung disease.2) On the other hand, STING ki;Tnfr1-/- (lacking TNFR1) showed the similar modest rescue of the CD8^+^ T and CD4^+^ T cells in blood compared to the WT C57BL/6 (BL6) but not with STING ki;Tnfr2-/- (lacking TNFR2). STING ki;Tnfr1-/-, STING ki;Tnfr2-/- and STING ki;Tnfr1/2-/- had modest rescue of thymic cell count and reduced spleen cell count (reduced splenomegaly). Along with the rescued thymic content and reduced splenomegaly, genetic ablation of TNF signalling (STING ki;Tnfr1-/-) also prevented manifestation of severe inflammatory lung disease.3) To investigate the role of lung endothelial cells in the development of interstitial lung disease, primary murine lung endothelial cells from STING WT, STING ki and STING WT;Tnfr1/2-/- and STING ki;Tnfr1/2-/- mice were isolated and bulk RNAseq was performed. This showed decreased level of several proinflammatory cytokines (e.g. Tnf, Il1b) and chemokines (e.g. Cxcl1, Cxcl2, Cxcl9, Cxcl10, Ccl2, Ccl3 and Ccl4) in STING ki mice lacking TNFR1/2 compared to STING ki mice.4) Neutrophils were isolated from bone marrow and were added to cultured primary lung endothelial cell monolayers. The experiments demonstrated that the attachment and transmigration of neutrophil cells were dependent on expression of STING gain-of-function mutation in endothelial cells.A few points require clarification before publication of this study.1. Tnfr1-/-, Tnfr2-/- and Tnfr1/2-/- did not show any statistical significant improvement of thymic cell count in STING ki mice. As such, the statement in the conclusion/summary section of discussion regarding Tnfr1 can restore thymocyte numbers should be toned-down.

We thank the reviewer for his/her time and for the constructive comments. Below please find our detailed responses to your points.

Thank you for this suggestion. In Figure 4 E, we demonstrated that knock out of TNFR1 leads to increasing of SP CD8 thymocyte count and partially of SP CD4 thymocyte count (Figure 4 D). In agreement with this suggestion, we marked this subpopulation of thymocytes in the discussion and summary section, see actual line 684 and see actual line 794.

2. The section on Neuroinflammation and neurodegeneration and dependency of TNFR1/2 signaling is very currently difficult to follow (based on how the data are presented in figures and text). This section requires to be re-written for clarity.

Thank you for this suggestion. We re-wrote this section, see line 472 – 499.

“Neuroinflammation and neurodegeneration in dependency of TNFR1/2 signaling

The extent of inflammation in mouse brain resulting from constitutive activation of STING N153S was reported by quantifying the density of Iba1-positive microglia (Figure 5 A). Consistent with our previous findings (Szego et al., 2022), the density of Iba1-positive microglia in the *substantia nigra* was higher in STING ki;BL6 mice than in STING WT mice (Figure 5 B). TNFR deficiency did not affect neuroinflammation because there was no significant difference between the density of Iba1-positive microglia between STING ki;BL6 mice and STING ki;*Tnfr1/2^-/-^* mice (Figure 5 B). This suggests that the TNF pathway is not required for STING-induced microglia activation in the *substantia nigra*.

In addition, we measured the extent of STING-induced astrogliosis by quantifying the density of GFAP-positive cells (Figure 5 A). Consistent with our previous findings, the density of GFAP-positive astroglia was higher in STING ki than in STING WT mice (Figure 5C). Yet, as for microglia, there was no significant difference between the density of GFAP-positive astroglia between STING ki;BL6 mice and STING ki;*Tnfr1/2^-/-^* mice (Figure 5 C), suggesting that the TNF pathway is not required for STING-induced astrogliosis in the *substantia nigra*.

Finally, we measured the extent of STING-induced neurodegeneration by quantifying the density of TH-positive dopaminergic neurons in the *substantia nigra* (Figure 5A)*.* As in our previous findings, the density of TH-positive neurons was lower in STING ki;BL6 mice than in STING WT mice (Figure 5 D). The density of TH-positive neurons in the *substantia nigra* of STING ki;*Tnfr1/2^-/-^* mice was higher than the density of TH-positive neurons in the *substantia nigra* of STING ki;BL6 mice (Figure 5 D), suggesting that the STING-induced degeneration of TH-positive neurons was blunted in *Tnfr1/2^-/-^* mice and that TNFR1/2 are involved in the STING-induced degeneration of dopaminergic neurons.

Hence, there is a discrepancy between STING-induced effects on glial cells as opposed to STING-induced effects on neurons. The dependence of STING-induced neurodegeneration but not glial response on TNFR1/2 suggests that the STING-induced degeneration of dopaminergic neurons is not a direct consequence of microglia or astroglia activation. This is consistent with the emerging concept of a neuron-specific inflammatory response (Welikovitch et al., 2020).”

The powerful use of in vivo genetic KO models and TNF inhibitor makes this study a valuable contribution to the field – helping further decipher the importance of the NF-ΚB/TNF branch of STING in SAVI (knowledge gap). The audience for this work would be specialised to STING biology and potential clinical treatments of SAVI.Our expertise is in nucleic acids sensing (such as STING) and auto-immunity.

[Editors’ note: what follows is the authors’ response to the second round of review.]

Based on the previous reviews and the revisions, the manuscript has been improved but there are some remaining issues that need to be addressed, as outlined below:We have read briefly the manuscript and accompanying comments, some of which are fair and some are not. We think this is a reasonable effort to understand the role of TNFRs in the pathogenesis of STING-associated vasculopathy. A significant amount of the disease observed is secondary to TNFRI activation and partially on TNFRII. Our greatest concern here relates to the use of Infliximab (an anti-human TNF antibody) rather than etanercept. There are studies which question the specificity of infliximab in the murine system (see for example, Anti‐inflammatory effects of infliximab in mice are independent of tumour necrosis factor α neutralization. Clin Exp Immunol. 2017 Feb; 187(2): 225-233. doi: 10.1111/cei.12872). Etanercept is known to bind to and inactivate both human and murine TNF. Although Infliximab has been used in the literature, its effects have been questioned over time and most recent studies preferentially use etanercept as a tool to block TNF-a in mice. The authors do present some results using etanercept (supplementary material). Unless the authors can show convincingly that murine TNF-a is being blocked in their system by Infliximab, this whole part of the study needs to be removed.

We thank you and the reviewers for your time and for the constructive comments which have helped us to substantially improve the manuscript. We have extensively revised the manuscript and omitted experimental data.